# A Multi-Control Strategy to Achieve Autonomous Field Operation

Cyrille Pierre [1,*], Roland Lenain [1], Jean Laneurit [1] and Vincent Rousseau [2]

1   INRAE, F-63170 Aubière, France
2   Sabi-Agri, F-63360 Saint-Beauzire, France
*   Correspondence: cyrille.pierre@inrae.fr

**Abstract:** Nowadays, there are several methods of controlling a robot depending on the type of agricultural environment in which it operates. In order to perform a complete agricultural task, this paper proposes a switching strategy between several perception/control approaches, allowing us to select the most appropriate one at any given time. This strategy is presented using an electrical tractor and three control approaches we have developed: path tracking, edge following and furrow pursuing. The effectiveness of the proposed development is tested through full-scale experiments in realistic field environments, performing autonomous navigation and weeding operations in an orchard and an open field. The commutation strategy allows us to select behavior depending on the context, with a good robustness with respect to different sizes of crops (maize and bean). The accuracy stays within ten centimeters, allowing us to expect the use of robots to help with the development of agroecological principles.

**Keywords:** robot control; path tracking; field navigation; plant tracking; finite state machine





## 1. Introduction

Climate change has become an emergent subject to be addressed without delay in human activities, as numerous studies [1] show the necessity of reducing environmental impacts. While all sectors of activity must make considerable efforts, agriculture is particularly important in the fight against global warming [2]. Current agriculture practices indeed use a consequent amount of pesticides and chemicals that have a non-negligible impact on the environment. Even though such practices have allowed us to increase yield and production levels to feed a growing worldwide population [3], it has become crucial to develop alternative practices, enabling us to avoid the use of chemicals as well as soil degradation. In order to tackle environment-friendly production, several new agricultural practices have been proposed, such as precision agriculture (PA) or organic farming [4]. Such examples of new routes can be gathered into the concept of agroecology [5], which can be viewed as the study of the ecological processes applied to agricultural production systems. It consists in mixing different species and crops in the same area in order to preserve soil and protect vegetation without using chemicals. These new ways for farming nevertheless require frequent treatment and a regular monitoring of cultures. Agricultural operations are moreover more complex, as several kinds of crops have to be managed separately in the same area and with different seasonalities. As a result, the rise of agroecology needs increased manpower, despite the work to be achieved appearing to be harsh and painful. Farming jobs then appear to be less and less attractive, leading to a lack of manpower [6]. The (r)evolution of agricultural practices then requires us to develop new tools, working with a high level of precision, without needing the use of hard human work.

As it has been observed in industry, robotics may appear as a promising solution to achieve these accurate, difficult and repetitive operations in the field [7]. Several autonomous platforms have then been marketed in the area of agriculture to address the

problem of work penibility and changing practices as well [8]. The use of robots in such natural environments is nevertheless not straightforward from an industrial background, as the area is low-structured, with a changing context and task to be achieved, deeply impacting the perception and the control of mobile robots acting off-road [9]. For the autonomous navigation functionality, a lot of efforts have been made to preserve the accuracy of robots in accordance with agricultural requirements (a few centimeters) in different contexts (absolute path tracking, row following, etc.). Robust approaches are often viewed as a solution, assimilating the variation in grip conditions as a bounded perturbation to be rejected [10]. Such approaches are successfully applied in the framework of off-road path tracking [11] but are hardly usable in agriculture due to the chattering effect, which may influence the quality of work achieved by an implement. The complexity of modeling mobile robots dynamics and dealing with perception conditions changes, bringing us to consider the use of learning-based approaches, such as in [12] for control or in [13] for row detection. Nevertheless, such approaches require, on the one hand, obtaining an important collection of data or having a realistic simulation framework. On the other hand, such points of view lose the deterministic properties allowing us to warrant the stability of the robots' behavior.

Adaptive approaches bring a variation in robot dynamics, especially in the framework of path tracking. Ref. [14] proposes an adaptive approach based on optimal control, while, in our previous work [15], a dynamic observer is designed to achieve an accurate trajectory following on different kinds of soil, with different speeds. Such an adaptation, coupled with predictive layers [16], allows us to preserve the accuracy within a few centimeters despite harsh dynamics and terrain variations. Nevertheless, they are often limited to one application such as path following using an absolute and accurate localization. A key challenge in agriculture robotics is not limited to autonomous navigation, as robots need to interact with vegetation and soil. As a result, it also has to reference its position with respect to crops, when it is in the field, while navigating without visible landmarks, when achieving a half-turn or while traveling between the farm and the field. The achievement of a complete agricultural task by a robot then requires several types of perception and algorithms, depending on the context, the task to be performed and the actual situation.

As a consequence, the adaptation of the robot must be extended to commutation between several behaviors (such as trajectory following, target tracking, structure following, etc.) in order to address the complexity of agricultural operations [17]. A natural way to manage the successive task of course lies in off-line planning, allowing us to define the general robotic mission [18]. If this initial planning constitutes an important element of allowing complex scenarios to be achieved, it first requires a semantic representation of the environment [19], not always easy to obtain. Secondly, the uncertainty and variability about the crops' growth or the probability of encountering unexpected situations requires the robot to adapt itself to the situation. As a result, on-line switching between several modes of control must be developed. Behavior-based control approaches such as those proposed in [20] for off-road navigation also constitute a possible way. Nevertheless, the co-existence of several control actions, potentially in contradiction, may lead to inaccuracies or oscillations. In the framework of an agricultural task, this may quickly lead to crop destruction, depending on the situation and the on-boarded implement.

In this paper, we propose a global strategy allowing us to switch between different perception and control approaches on an electrical tractor that is foreseen to be automated, depicted in Figure 1. Three main behaviors are described in the framework of this paper: *path tracking*, *edge following* and *furrow pursuing*. The proposed approach consists in automatically switching between the different behaviors depending on the situation, i.e, the result of Lidar detection with respect to an expected shape of crops. The algorithm takes part in a predefined planned trajectory to be followed if no crops are detected. Three criteria are proposed to achieve the selection depending on the detection relevance. Such a strategy allows us to select the best behavior without requiring the use of a semantic map or a huge robotic mission planning description. The effectiveness of the proposed

approach is evaluated in full-scale experiments in a three-environment testbed. The first simulates an autonomous navigation in a vineyard, switching between *path tracking* and *edge following*. The second and the third permit us to achieve the autonomous weeding of a field with different kinds of crops (maize and beans) that have not been sowed in straight lines. This testbed has been achieved in the framework of the "Acre" challenge of the METRICS project [21]. Results show the ability to satisfactorily select the expected behavior, allowing us to keep an accuracy of within few centimeters, despite changing vegetation. This validates the ability to autonomously achieve a highly accurate and complete field operation, being positioned with respect to the crops when needed.

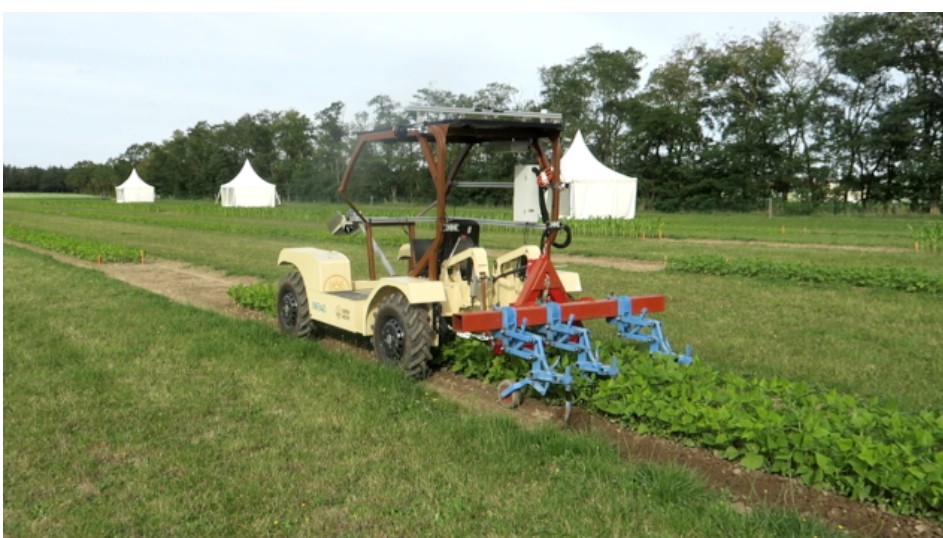

**Figure 1.** SabiAgri electrical and automatized tractor used for full-scale experiments.

The paper is composed as follows. First, the global approach is described and the different behaviors are briefly exposed. It permits us to obtain three possible approaches to select depending on the situation. In the second part, the switching strategy is detailed, using the criteria proposed in the paper. It allows us to make a decision on the adapted behavior depending on the robot state and environment perception. The experimental results are then discussed in the third section. The experiment is based on two use cases: orchard navigation and open field weeding treatment. The results allow us to conclude on the effectiveness of the proposed algorithm to achieve complex agricultural tasks due to a fully autonomous robot.

## 2. Definition of Elementary Robotics Behaviors

In this paper, a global adaptation strategy to select the behavior of an agricultural robot is proposed in order to achieve complete operations depending on the encountered situation. As has been mentioned previously, we consider that the diversity of robotics tasks to be addressed in a field treatment cannot be handled by a single perception-control approach, as the robot must move in the farm, travel up to the field, track and interact with some rows of plants, make maneuvers, etc. To allow a high level of versatility for farming robots, an algorithm dedicated to switching between different elementary behaviors is proposed. Before detailing this architecture, let us first describe the three elementary approaches that are considered in this paper, allowing us to realize farming operations:

**Path tracking.** This first approach consists in following a known trajectory (planned or previously learned) in an absolute framework. This permits us to have an absolute autonomous navigation framework, without considering any visible reference. This behavior can be used to achieve maneuvers, travel between fields or move through the farm. It requires access to GNSS information or an absolute localization system.

**Edge following.** This second behavior is dedicated to moving relatively to an existing structure using a horizontal 2D Lidar. It selects impact points to derive a relative trajectory to be followed with an offset. This is used to follow structures in indoor environments (such as walls or fences in farm buildings) or in fields (such as tree rows in orchards or vineyards).

**Furrow pursuing.** This third approach consists in finding an expected shape on the ground using an inclined 2D Lidar. The objective is to recognize ground shape (such as footprint or crops) in the Lidar framework. The accumulation of detected points through the robot allows us to define a local trajectory to be followed.

The behavior *path tracking* is considered in the following as a default behavior and is used when no specialized behavior is available. The two others are considered as specialized behaviors and are used when the elements to be observed are detected. They are selected in priority when detecting plants, as the objective of agricultural works is to move with respect to plants. The aim of the algorithm is to favor the specialized behaviors while maintaining consistent control. Before describing the selection algorithm in Section 3, let us detail further the three elementary behaviors here before mentioned.

### 2.1. Path Tracking

The first elementary behavior investigated in this paper relies on the path following of an absolute trajectory. To this aim, let us consider the robot moving on a plane with a longitudinal symmetric axis with respect to the middle of the rear axle. Equipped with these assumptions, the robot can be viewed as a bicycle model such as the one depicted in Figure 2. The notation used in the following is reported in this figure. The point to be controlled is the middle of the rear axle $O$. The objective is to make it converge to the reference trajectory $\Gamma$. While the velocity is supposed to be controlled independently, one considers the speed vector $v$ of the point $O$ has a known variable. The position and the velocity of $O$ in the world frame are computed using an extended Kalman filter that merges the odometry of the robot, an inertial measurement unit (IMU) and a real-time kinematic (RTK) global navigation satellite system (GNSS). The robot is modeled in a Frénet frame, situated at the closest point from $O$ belonging to $\Gamma$. At this point, one can define:

-   $s$, which is the curvilinear abscissa;
-   $y$, the tracking error;
-   $\tilde{\theta}$, the angular deviation, which is defined by the difference between the robot heading and the tangent of the trajectory at point $s$.

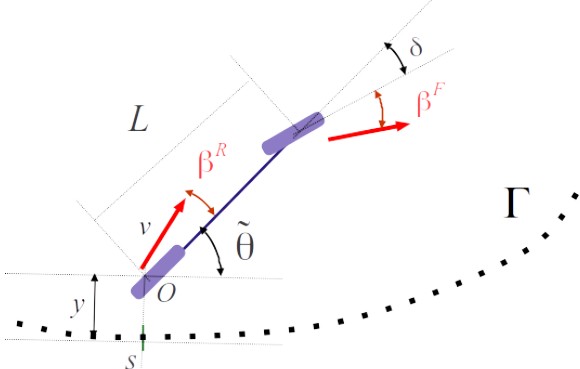

**Figure 2.** Extended kinemtic model of the robot for path following purpose.

In order to account for the bad grip conditions encountered in the off-road context, we also define two variables representative of the fact that rolling without sliding conditions is not satisfied. The two sideslip angles $\beta^F$ and $\beta^R$ are then introduced for, respectively, the front and the rear wheels. These variables are representative of the difference between each wheel orientation and the actual speed vector one. These two sideslip angles cannot

be directly measured. As a result, an observer is used as detailed in [22], allowing us to satisfactorily estimate these sideslip angles. Using these notations, one can derive the extended kinematic model giving the evolution of the state variables $[y, \tilde{\theta}]^T$ with respect to curvilinear abscissa $s$:

$$\begin{cases} y' &= (1 - yc(s)) \tan(\tilde{\theta}_2) \\ \tilde{\theta}' &= u_\theta \frac{1 - yc(s)}{v \cos(\tilde{\theta}_2)} - c(s) \end{cases} \tag{1}$$

with $\tilde{\theta}_2 = \tilde{\theta} + \beta_R$ and $u_\theta = v \cos \beta^R \frac{\tan(\delta^F + \beta^F) - \tan \beta^R}{L}$ the yaw rate of the robot. $c(s)$ is the curvature of the reference point at $s$ and $L$ is the robot wheelbase. Variable $x' = \frac{dx}{ds}$ denotes the derivative with respect to the curvilinear abscissa, allowing us to obtain a convergence in distance, instead of time, in order to have a robot control independent from the speed.

Once the state vector $[y, \tilde{\theta}]$ is known, the objective of *path tracking* is to control the steering angle $\delta$. This is achieved due to a backstepping approach composed of two steps.

### 2.1.1. Step 1: Optimal Orientation Computation

As the objective is to ensure the convergence of the tracking error $y$ to zero, one can define a differential equation imposing this behavior, such as:

$$y' = k_y y \tag{2}$$

with $k_y$ a gain defining the exponential convergence distance. By injecting the expression in (1) into this condition, one can obtain the target orientation:

$$\tilde{\theta}^{Obj} = \arctan \left\{ \left( \frac{k_y y}{1 - yc(s)} \right) \right\} - \beta^R \tag{3}$$

If the robot relative orientation $\tilde{\theta}$ is equal to $\tilde{\theta}^{Obj}$, then the differential equation $y' = k_y y$ is ensured and the lateral error will converge to zero. The second step then consists in finding the steering angle imposing the convergence $\tilde{\theta} \to \tilde{\theta}^{Obj}$.

### 2.1.2. Step 2: Steering Angle Control

As has been mentioned, the second step consists in ensuring the convergence of the angular deviation $\tilde{\theta}$ to the target one $\tilde{\theta}^{Obj}$. Let us define the error $e_{\tilde{\theta}} = \tilde{\theta} - \tilde{\theta}^{Obj}$. A condition to ensure this convergence is to impose:

$$e'_{\tilde{\theta}} = k_\theta e_{\tilde{\theta}} \tag{4}$$

Let us substitute into $e'_{\tilde{\theta}}$ the second row of the kinematic model in (1) and consider that the variations of target orientation can be neglected with respect to the angular deviation control. One can write:

$$u_\theta^{Obj} = \frac{[k_\theta e_{\tilde{\theta}} + c(s)] v \cos \tilde{\theta}_2}{1 - yc(s)} \tag{5}$$

$u_\theta^{Obj}$ then constitutes the expression of the yaw rate to be imposed to the robot for ensuring the condition in (4). By considering the expression of $u_\theta$, one can derive directly the control law expression for the steering angle:

$$\delta = \arctan \left\{ \frac{L[k_\theta e_{\tilde{\theta}} + c(s)]}{1 - yc(s)} + \tan \beta^R \right\} - \beta^F \tag{6}$$

This control expression constitutes the steering angle $\delta$ to be sent to the robot to ensure the expression in (4) is true and finally the convergence of the tracking error $y$ goes to zero. A condition to be met is that the convergence distance for the angular deviation is shorter than the one imposed for the lateral error. This can be ensured by imposing $k_\theta \gg k_y$. In addition to this control expression, it has been proposed in [22] to apply a predictive

layer on the curvature servoing, allowing us to compensate for low-level delay and settling time. In practice, an anticipation effect can also be obtained by feeding forward the control law with future curvature $c(s + s_H)$, considering $s_H$ as a distance of prediction. As the robot moves at low speed, in this paper, the second solution is here applied considering that settling time of the actuator is equal to $T$. One can then finally rewrite the control expression as:

$$\delta = \arctan\left\{\frac{L\left[k_\theta e_{\tilde{\theta}} + c(s + vT)\right]}{1 - yc(s)} + \tan\beta^R\right\} - \beta^F \tag{7}$$

The same control is then applied in the forthcoming strategy. Nevertheless, the lack of accuracy and noises obtained with other sensors does not permit us to satisfactorily observe sideslip angles. As a result in other behaviors, sideslip angles $\beta^F$ and $\beta^R$ are neglected and set to zero in the control expression in (7).

### 2.2. Edge Following

The aim of *edge following* is to follow a linear object such as a wall, a row of posts or a row of plants. The shape of the object to follow is detected using a horizontal 2D Lidar (Figure 3a). The Lidar is able to perceive a set of range measurements for different angles at the front of the robot. These measurements are converted into 2D points on a plane parallel to the ground (Figure 3b). When the robot is close to the objects to follow, the points of the Lidar form one or several clusters on the left, on the right or on both sides of the robot. It is then possible to build a trajectory from the shape of the edge of these clusters.

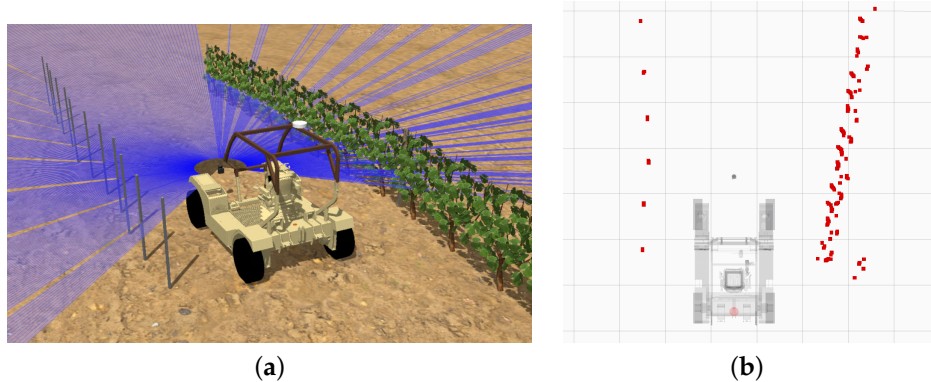

(**a**)        (**b**)

**Figure 3.** Placement of the horizontal 2D Lidar on the robot. The measurements of the sensor are converted into 2D points in the robot frame. (**a**) Visualization of the Lidar rays (in blue); (**b**) 2D projection of the Lidar points.

The algorithm used to detect the shape is based on a circle of radius $r$ that rolls along the cluster. It is presented in Algorithm 1. If the cluster is on the right of the robot, the circle rolls on the left of the cluster, and if the cluster is on the left of the robot, the circle rolls on the right of the cluster. This allows us to build a chain from selected Lidar points so that the robot can traverse the observed area while following the object. The first point of the chain is selected by initializing the circle at the rear of the robot $(x_r, y_r)^\top$ and by translating it along an axis perpendicular to the robot's forward axis. The translation is conducted progressively until a point of the Lidar touches the circle (Figure 4a). This point is then the first in the chain. To avoid initializing the chain from a point on the wrong side of the robot, we only consider the points that touch a semicircle oriented in the direction of the translation. The next point of the chain is found by progressively rotating the circle around the previous point of the chain until a new point of the Lidar touches the circle (Figure 4b). This new point is then added to the chain. The rotation of the circle is repeated until a rotation angle $\theta_{\max}$ limit or a chain length $l_{\max}$ limit is reached (Figure 4c).

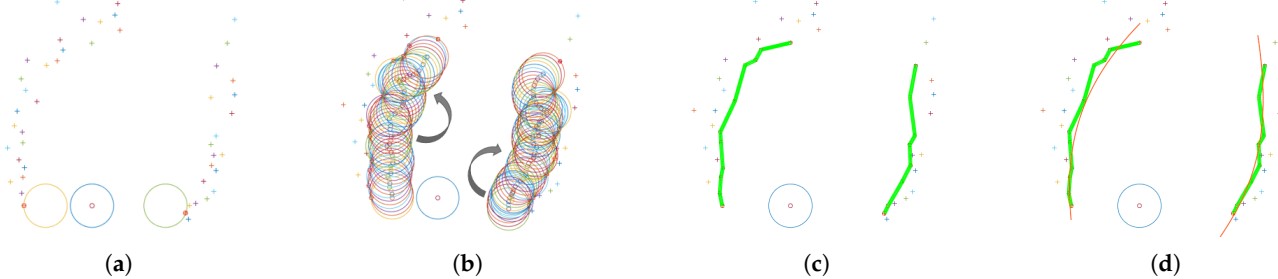

(a)        (b)        (c)        (d)

**Figure 4.** Steps of the *edge* detection algorithm applied on both sides of the robot. The center of the blue circle at the bottom of each figure corresponds to the position of the robot. (**a**) shows the translated circles and the initial points of the chains. (**b**) shows the rotation of the circles along each previous point of the chain. (**c**) shows the obtained chain for the left and the right side. (**d**) shows the polynomial regressions of the chains.

---

**Algorithm 1:** Building the edge chain

---

**Data:** $\mathcal{P}$, the list of Lidar 2D points

      $s$, the side (left or right) of the robot where the chain will be built

**Result:** $\mathcal{C}$, the list of points of the chain

**begin**

    $\mathcal{C} \leftarrow \varnothing$

    $\mathbf{c} \leftarrow (x_r, y_r)^\top$, the current center of the circle

    $\mathcal{D}_{\mathbf{c},r}$, the disk of center $\mathbf{c}$ and radius $r$

    **while** $\mathcal{P} \cap \mathcal{D}_{\mathbf{c},r} = \varnothing$ **do**

        $\mathbf{c} \leftarrow \mathbf{c} + \mathbf{t}_s$, where $\mathbf{t}_s$ is a short translation vector in the direction of $s$

    $\mathbf{p} \leftarrow \underset{\mathbf{x} \in \mathcal{P} \cap \mathcal{D}_{\mathbf{c},r}}{\arg\min} (\mathbf{c} - \mathbf{x})^2$, the current pivot point

    $\mathcal{C} \leftarrow \mathcal{C} \cup \{\mathbf{p}\}$

    $\theta \leftarrow 0$, the rotation angle around the pivot point $\mathbf{p}$

    **while** $\mathrm{length}(\mathcal{C}) < l_{\max}$   *and*   $\theta < \theta_{\max}$ **do**

        **while** $\mathcal{D}_{\mathbf{c},r} \cap \mathcal{P} \setminus \{\mathbf{p}\} = \varnothing$   *and*   $\theta < \theta_{\max}$ **do**

            $\theta \leftarrow \theta + \theta_s$, where $\theta_s$ is a small signed angle that depends on $s$

            $\mathbf{c} \leftarrow \mathbf{p} + \mathrm{rot}(\theta_s)(\mathbf{c} - \mathbf{p})$, where $\mathrm{rot}(\theta_s)$ is a rotation matrix of angle $\theta_s$

        **if** $\theta < \theta_{\max}$ **then**

            $\mathbf{p} \leftarrow \underset{\mathcal{D}_{\mathbf{c},r} \cap \mathcal{P} \setminus \{\mathbf{p}\}}{\arg\min} (\mathbf{c} - \mathbf{x})^2$

            $\mathcal{C} \leftarrow \mathcal{C} \cup \{\mathbf{p}\}$

            $\theta \leftarrow 0$

---

The generated chain of Lidar points $\mathcal{C}$ is not directly used as a trajectory to follow. It is first approximated as a parabola by computing a linear regression of a polynomial of degree two on the chain (Figure 4). The objective is then to find $(a, b, c)$ such that it solves the equation

$$a\, p_{x,i}^2 + b\, p_{x,i} + c = p_{y,i} \tag{8}$$

for every point $(p_{x,i}, p_{y,i}) \in \mathcal{C}$. The result is given by

$$\begin{bmatrix} a & b & c \end{bmatrix}^\top = (\mathbf{X}^\top \mathbf{X})^{-1} \mathbf{X}^\top \mathbf{Y} \tag{9}$$

where

$$
\mathbf{X} = \begin{bmatrix} p_{x,0}^2 & p_{x,1}^2 & \cdots & p_{x,n}^2 \\ p_{x,0} & p_{x,1} & \cdots & p_{x,n} \\ 1 & 1 & \cdots & 1 \end{bmatrix}^\top
$$
$$
\mathbf{Y} = \begin{bmatrix} p_{y,0} & p_{y,1} & \cdots & p_{y,n} \end{bmatrix}^\top \tag{10}
$$
$$
n = |\mathcal{C}|
$$

It is then possible to compute the lateral error $y$ and angular error $\tilde{\theta}$ of the robot position $(x_r, y_r)$ to the parabola:

$$
\begin{aligned}
y &= a\, x_r^2 + b\, x_r + c \\
\tilde{\theta} &= \arctan(2\, a\ x_r + b)
\end{aligned} \tag{11}
$$

These errors are then used as input parameters for the backstepping robot control approach presented in Section 2.1. However, this version of the control law does not include an estimation of sideslip angles.

*2.3. Furrow Pursuing*

The aim of *furrow pursuing* is to follow a linear object such as wheelprints, one or two rows of small plants or a trellis of vine. This approach also uses a 2D Lidar to perceive the elements to follow but, contrary to *edge following*, the Lidar is inclined to look at the ground. In this configuration, the sensor cannot make a complete observation of the trajectory to follow because it can only perceive a slice of the environment at a given time. The trajectory is deduced by the accumulation of data obtained from several Lidar observations which are geometrically repositioned due to the odometry of the robot.

The first step of the detection algorithm is to convert the measurements of the 2D Lidar into 3D points in the robot frame (Figure 5a). The axes $(x, y, z)$ of the robot frame correspond to the *forward*, *left* and *up* axes, respectively. It is not necessary to take the full angular range of the Lidar since we are only interested in the ground. Therefore, the detection algorithm is limited to a small angular range around the $x$ axis of the robot frame. The next step is to find one or more occurrences of the shape of the feature to follow in the Lidar scan. To do this, we compute a correlation function between the $z$ coordinates of the Lidar points and a reference model chosen specifically for the shape we want to track. Let $z_a[k]$ be a discrete function containing the $z$ coordinate of the point $k$ of the restricted Lidar scan and $z_r[k]$ the $z$ coordinate of the point $k$ of the reference model. The model corresponds to a rectangular function for which we configure the height $h_{\text{ref}}$ and the width $w_{\text{ref}}$ (in number of Lidar rays) to match the dimension of the element to follow. It can be expressed as

$$
z_r[k] = \begin{cases} h_{\text{ref}} & \text{if } k \in [-\frac{w_{\text{ref}}}{2}, \frac{w_{\text{ref}}}{2}] \\ 0 & \text{else} \end{cases}. \tag{12}
$$

The correlation between the Lidar scan and the reference model corresponds to

$$
c[k] = \frac{1}{\sqrt{A\,R}} \sum_{i \in \mathbb{N}} z_r[k]\, z_a[k+i] \tag{13}
$$

with

$$
A = \sum_{i \in \mathbb{N}} (z_a[k])^2, \quad R = \sum_{i \in \mathbb{N}} (z_r[k])^2.
$$

In order to obtain the set of matches between the reference model and the Lidar measurement, the algorithm extracts a set of local maxima $\mathcal{M}$ in the correlation curve $c[k]$ if these maxima are greater than a correlation threshold $c_t$ (Figure 5b). The rest of the algorithm depends on the number of rows that the robot has to follow.

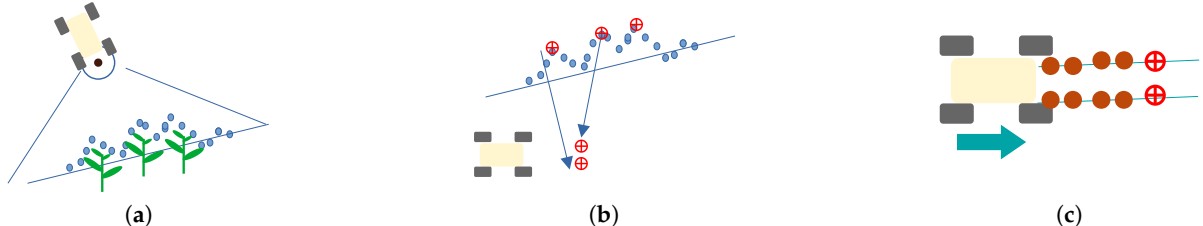

**Figure 5.** Steps of the *furrow* detection algorithm. (**a**) Measurement; (**b**) correlation; (**c**) updated positions.

### 2.3.1. The Robot Is Configured to Track One Row

The selected maximum point $\mathbf{m}_t = (m_{x,t}, m_{y,t})^\top \in \mathcal{M}$ at a given time $t$ corresponds to the closest one of the previous measurement $\mathbf{m}_{t-1}$ or the closest to the robot position in the $y$ axis.

$$\mathbf{m}_t = \begin{cases} \underset{\mathbf{m} \in \mathcal{M}}{\arg\max}(m_y) & \text{if } t = 0 \\ \underset{\mathbf{m} \in \mathcal{M}}{\arg\max}(m_y - m_{y,t-1}) & \text{else} \end{cases} \tag{14}$$

The previously selected points are grouped in a set $\mathcal{P}_t$ that is updated at each instant $t$ in order to remain accurate with respect to the movement of the robot. This movement is computed using robot odometry measurements and the evolution model of the robot. Points that are far from the robot are removed to keep the trajectory estimation local. The trajectory is obtained by applying a linear regression from the points $\mathbf{p}_i = (p_{x,i}, p_{y,i})^\top \in \mathcal{P}_t$. The equation of the regression is

$$a\, p_{x,i} + b = p_{y,i} \tag{15}$$

where $(a, b)$ are the coefficients to solve. These coefficients allow us to compute the lateral error $y$ and angular error $\tilde{\theta}$ from the robot position $(x_r, y_r)$ to the trajectory. Because the points are expressed in the robot frame, these errors can be defined as

$$y = \frac{b}{\sqrt{a^2 + 1}} \tag{16}$$
$$\tilde{\theta} = \arctan(a)$$

These errors are then used as input for the backstepping robot control approach presented in Section 2.1. However, this version of the control law does not include an estimation of sideslip angles or the predictive layer.

### 2.3.2. The Robot Is Configured to Track Two Rows

If the algorithm tracks two rows, the maxima $\mathcal{M}$ are grouped into a set of pairs of Lidar points $\mathcal{P}$, keeping only those where the distance in the robot's $y$ axis are within a chosen interval $\mathcal{D} = [d_{\min}, d_{\max}]$.

$$\mathcal{P} = \{(\mathbf{m}_1, \mathbf{m}_2) \mid \|\mathbf{m}_1 - \mathbf{m}_2\| \in \mathcal{D},\ (\mathbf{m}_1, \mathbf{m}_2) \in \mathcal{M}^2\} \tag{17}$$

To select the best pair $\mathbf{P}_t \in \mathcal{P}$ of the measurement $t$, it is necessary to define a function corresponding to the central point of a pair:

$$\mathrm{mid}(\mathbf{P}) = \frac{\mathbf{p}_1 + \mathbf{p}_2}{2}, \quad \text{with } \mathbf{P} = (\mathbf{p}_1, \mathbf{p}_2) \tag{18}$$

The selected pair $P_t$ is then obtained using the formula

$$\mathbf{P}_t = \begin{cases} \underset{\mathbf{P} \in \mathcal{P}}{\arg\min}\ \mathrm{mid}(\mathbf{P}) & \text{if } t = 0 \\ \underset{\mathbf{P} \in \mathcal{P}}{\arg\min}\ \mathrm{mid}(\mathbf{P}) - \mathrm{mid}(\mathbf{P}_{t-1}) & \text{else} \end{cases} \tag{19}$$

which corresponds to selecting the pair the closest to the robot on the $y$ axis or the closest to the previous selection ($\mathbf{P}_{t-1}$). The points of the $n$ last selected pairs are separated into two groups which correspond to the left points $\mathcal{L}_t$ and the right points $\mathcal{R}_t$. A linear regression similar to the one presented in Equation (15) is applied on each group $\mathcal{L}_t$ and $\mathcal{R}_t$ in order to obtain the coefficients ($a_l, b_l$) and ($a_r, b_r$) which describe a line for each row of plants. From these lines, it is then possible to compute the lateral and angular errors of the robot to each line ($y_l, \tilde{\theta}_l$) and ($y_r, \tilde{\theta}_r$) by using Equation (16). The error of the estimated trajectory is obtained by combining the errors of both lines:

$$
\begin{aligned}
y &= \frac{y_l + y_r}{2} \\
\tilde{\theta} &= \frac{\tilde{\theta}_l + \tilde{\theta}_r}{2}
\end{aligned}
\tag{20}
$$

These errors are then used as input for the backstepping robot control approach presented in Section 2.1. However, this version of the control law does not include an estimation of sideslip angles or the predictive layer.

## 3. Switching to the Appropriate Behavior

The behavior *path tracking* has the advantage of being fully defined for the realization of the agricultural task. It requires, however, that GNSS is available. In the case of the specialized behaviors *edge following* and *furrow pursuing*, the trajectory to follow is generated locally in real time and is not available everywhere because there are situations where there is nothing to track. This is the case, for example, of passages where the robot has to make a U-turn to take the next line of vegetation. The selection algorithm thus acts as a state machine and allows us to choose between two states for robot control: *path tracking* or one of the specialized behaviors.

Since there are several types of specialized behavior, there are several versions of this state machine. Figure 6 shows the components involved in the developed algorithm. In the function of the application, the specialization behavior can be *edge following* or *furrow pursuing*. Each behavior is able to provide a linear velocity and a steering angle to control the robot. The selection algorithm then only has to select which command to apply to the robot at a given time. The difficulty of this approach lies in the choice of the switching criterion between the different behaviors.

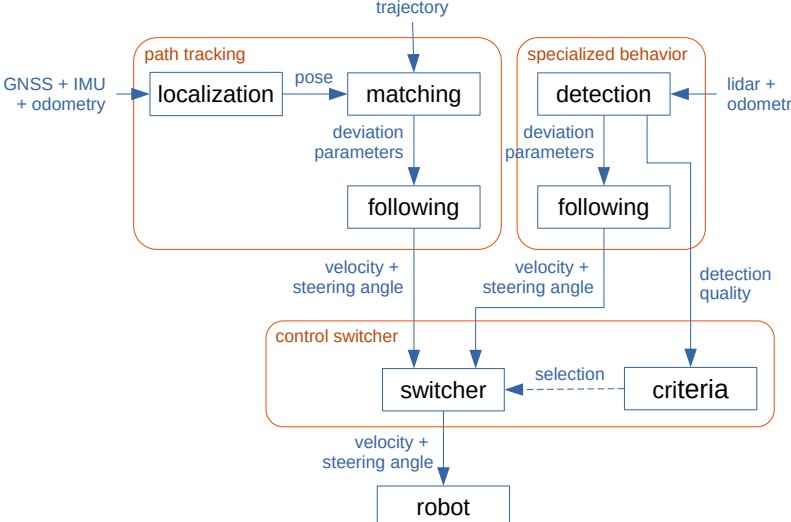

**Figure 6.** Description of the communication between the components of the selection algorithm.

The criterion is based on the quality of the detection of the specialized behavior. This quality corresponds to the ability of the detection algorithm to determine whether the

observed element corresponds to an element to follow. For *edge following*, the criterion is based on the following conditions:

- The length of the detected chain $\mathcal{C}$ is greater than a threshold $l_{min}$. If the chain is too short, then there is high probability that this is not an object to follow.
- The width of the chain $\mathcal{C}$ on the $y$ axis (in the robot frame) is lower than a threshold $w_{max}$. This avoids following an object that is not in the current direction of the robot.
- The standard deviation of the $y$ distance between the points of the chain and the parabola defined in Equation (8) is lower than a threshold $\sigma_{max}$. This allows us to quantify how complex the object to follow is. It is then possible to avoid following objects whose shapes do not correspond to the expected level of complexity.

These conditions can be written as

$$
\begin{cases}
\text{length}(\mathcal{C}) > l_{min} \\
\max_{(\mathbf{p}_1,\mathbf{p}_2) \in \mathcal{C}^2} (p_{y,1} - p_{y,2}) < w_{max} \\
\sum_{\mathbf{p} \in \mathcal{P}} (p_y - a\, p_x^2 + b\, p_x + c) < \sigma_{max}^2
\end{cases}
\tag{21}
$$

with $\mathbf{p}_i = (p_{x,i}, p_{y,i})^\top$.

For *furrow pursuing*, the criterion is based on the following conditions:

- The fact that the detection algorithm can find a pair of maxima in $\mathcal{P}$.
- The correlation level of each maximum of the selected pair at the indexes $k_l$ and $k_r$ is lower than a threshold $c_{max}$. This allows us to avoid false detection.

These conditions can be written as

$$
\begin{cases}
\mathcal{P} \neq \varnothing \\
c[k_l] < c_{max} \\
c[k_r] < c_{max}
\end{cases}
\tag{22}
$$

These criteria allow us to switch from *path tracking* to specialized behavior but also the reverse transition. The difference lies in the values used for the thresholds of the conditions. Using different values for these thresholds allows us to avoid quick and repetitive transitions between the two behaviors. We also use delays in the transitions to avoid a behavior remaining selected for too short a time.

## 4. Experiments

This section presents three experiments illustrating the algorithm for switching between specialized behaviors and *path tracking*. The robot used was an electrical tractor that weighed 800 kg, had a wheelbase of 2 m and had a track of 1.5 m. It was equipped with a GNSS Drotek F9P and an IMU Xsens MTi. The first experiment, which was performed at INRAE in Clermont-Ferrand (France, 63), used the *edge following* behavior to follow a hedge. The two other experiments were performed during an agricultural competition of the METRICS (Metrological Evaluation and Testing of Robots in International Competitions) project organized at Montoldre (France, 03). It is an application of the *furrow pursuing* behavior applied on an agricultural field composed of maize and bean plots.

### 4.1. Experiment 1: Edge Following

In this first experiment, the objective was to control the robot so that it followed a trajectory passing near a hedge. Figure 7a shows the complete trajectory and the part where the hedge is detectable and Figure 7b shows the robot during the experiment, at the beginning of its last right turn. The robot used mainly *path tracking* to follow this reference trajectory and could switch to *edge following* when the detection algorithm started to perceive the hedge. The reference trajectory was recorded manually and had no strong constraints on the distance to the hedge. The robot could therefore deviate from the reference trajectory when it was in *edge following* mode because the configured

distance, specified in Table 1, could not correspond to the distance between the hedge and the trajectory. Moreover, the robot could switch to *edge* when the trajectory was not yet completely parallel to the hedge because the hedge was already visible to the detection algorithm. In this configuration, the robot could detect the hedge if it was less than 3 m away, but the initial orientation of the robot did not allow us to directly switch to this behavior.

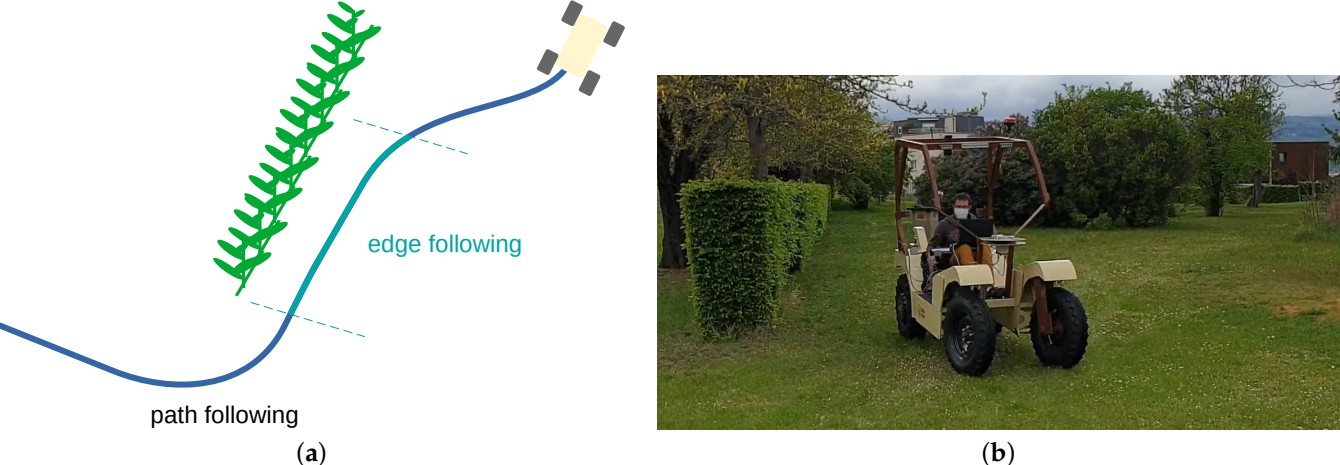

(**a**)                      (**b**)

**Figure 7.** Presentation of the experiment with *edge following*. (**a**) Scenario; (**b**) picture at the end of the hedge.

**Table 1.** Configuration of the experimentation with *edge following*.

| Parameter | Value | Unit |
|---|---:|---|
| **Configuration of *Path Tracking*** | | |
| desired linear speed | 0.8 | $\mathrm{m\,s^{-1}}$ |
| proportional gain of the control law | 0.7 | |
| time horizon of the prediction | 1 | s |
| **Configuration of *Edge Following*** | | |
| desired linear speed | 0.4 | $\mathrm{m\,s^{-1}}$ |
| desired lateral distance | 1.4 | m |
| radius of the rolling circle (for detection) | 1 | m |
| maximal length of the chain (for detection) | 3 | m |
| maximal distance to search an edge | 3 | m |
| proportional gain of the control law | 0.75 | |
| derivative gain of the control law | 0.45 | |
| **Configuration of the switching algorithm** | | |
| maximal standard deviation threshold to transit to *edge* ($\sigma_{\mathrm{edge}}$) | 0.15 | m |
| minimal standard deviation threshold to transit to *path* ($\sigma_{\mathrm{path}}$) | 0.5 | m |
| time to transit to *edge* | 0.4 | s |
| time to transit to *path* | 0.26 | s |

The results of the experiment are presented in Figure 8. Figure 8a shows the reference trajectory (in black) and the measured trajectory (in red and blue) of the robot computed using the localization step of *path tracking*. The trajectories are expressed in world frame with the $x$ axis oriented in the east direction and the $y$ axis in the north direction. Figure 8b shows the lateral error of the robot as a function of the curvilinear abscissa of the reference trajectory. The measured curves are red when the robot is in *path tracking* mode and blue when it uses *edge following*. At the beginning of the experiment (top left corner), the robot was not placed at the start point of the reference trajectory. We can see that the initial lateral

error was close to $-0.8$ m, but the robot gradually converged to the trajectory over the first 7 m.

At 9 m from the beginning of the trajectory, the robot switched to *edge following*. During this phase, the lateral error corresponded to the difference between the measured lateral distance to the trajectory generated from the hedge and the desired lateral distance. The initial error of this phase was $-0.4$ m because the robot switched to this behavior, but it was still far away. This phase also converged progressively with the displacement of the robot, but the error remained more noisy than during *path tracking* because the trajectory to follow was dynamic and depended on the shape of the hedge. We can notice in Figure 8a that the direction of the reference path was slightly different from the one followed by the robot. This was due to the fact that the recorded trajectory was not sufficiently parallel to the hedge. We can thus see that the use of a specialized behavior has an advantage on the quality of the tracking of the elements of the environment.

When the robot reached the end of the hedge, the algorithm switched back to *path tracking* and followed the end of the reference path. The initial lateral error of this third step was again important, but the robot converged on the rest of the trajectory. The control laws used in these behaviors had the particularity of giving progressive commands. This was also due to the fact that the speed of change in the steering angle was relatively slow on the robot we were using. This resulted in smooth transitions between the different behaviors.

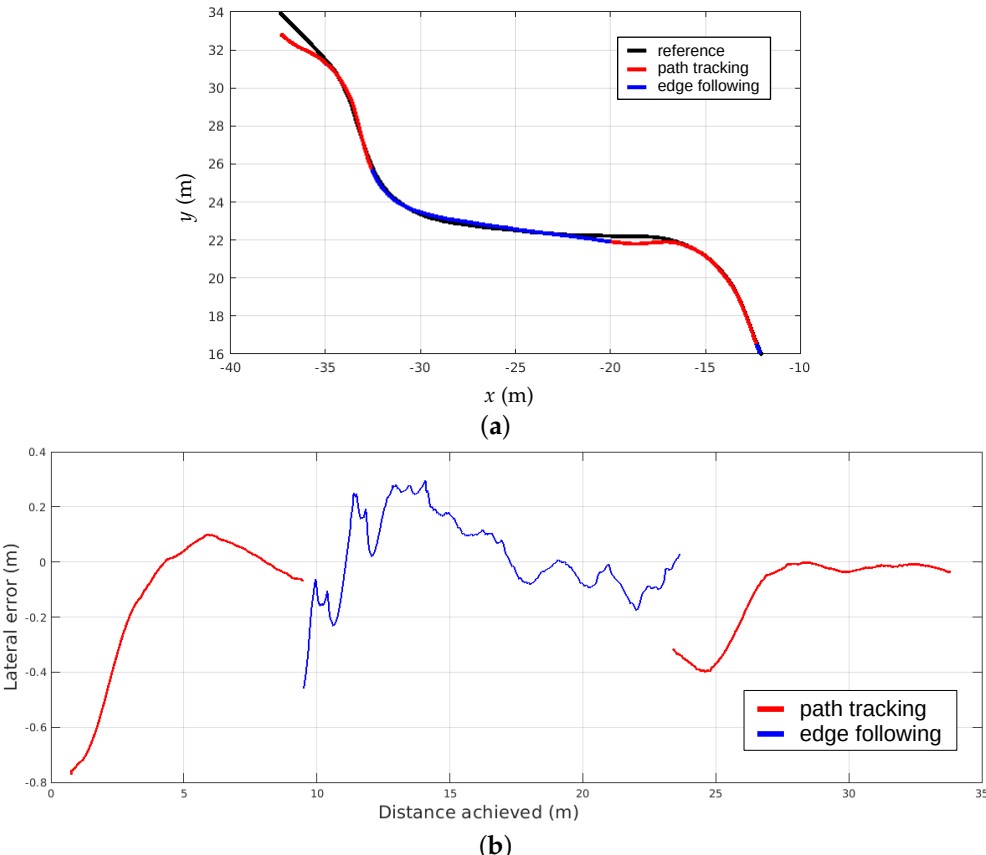

**Figure 8.** Results of experiment 1. (**a**) Absolute position $(x, y)$ of the robot. In red, the robot is in *path tracking* mode; in blue, it is in *edge following* mode. The black trajectory corresponds to reference trajectory to follow. (**b**) Lateral error in the Frénet frame of the local trajectory. For *path tracking* (in red), the local trajectory is the reference trajectory, and for *edge following* (in blue), it is built from the edge detection.

### 4.2. Experiment 2: Furrow Pursuing And U-Turn

The objective of this second experiment was to evaluate the developed switching algorithm between *path tracking* and *furrow pursuing*. It was conducted under conditions similar to a market garden environment. This experimental field was set up in the framework of the ACRE (Agri-food Competition for Robot Evaluation) organized by the METRICS project, whose objective is to evaluate different tasks such as weed control or field navigation [21]. The scenario of this experimentation, presented in Figure 9a, was realized on a training field composed of a maize plot and 3 bean plots. The maize plot consisted of two rows of plants 0.75 m apart and approximately 0.5 m high. For the bean plots, these were made up of three rows spaced 0.375 m apart and approximately 0.3 m high. In order to track both types of plants in the same run, the plant detection algorithm was configured to track two rows with a height and width corresponding to that of the smaller plants. The correlation calculation between the Lidar measurements and the plant model was permissive enough to handle beans and maize at the same time. The values of the configuration parameters are detailed in Table 2. In this scenario, all plots were parallel to each other. The idea was to cross each plot using the specialized behavior and switch to *path tracking* to perform the recorded U-turn trajectories during the transitions between plots.

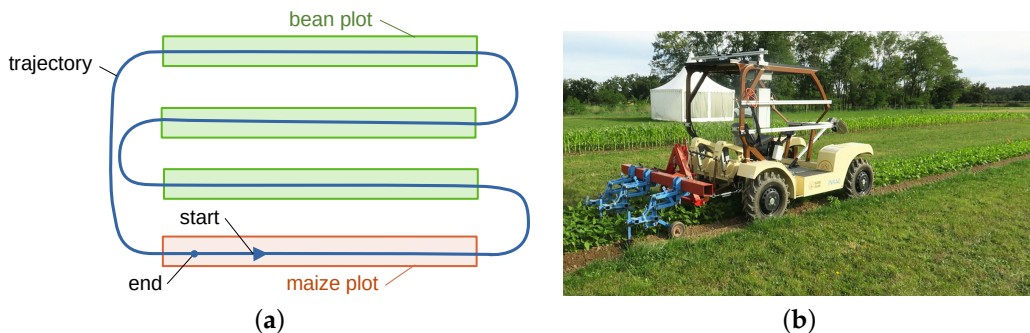

(**a**) (**b**)

**Figure 9.** Presentation of experiment 2. (**a**) Scenario; (**b**) picture of the robot in one of the bean plots.

**Table 2.** Configuration of the experimentation with *furrow pursuing*.

| Parameter | Value | Unit |
|---|---|---|
| **Configuration of *Path Tracking*** | | |
| desired linear speed | 0.8 | $m\,s^{-1}$ |
| proportional gain of the control law | 0.7 | |
| time horizon of the prediction | 1 | s |
| **Configuration of *furrow pursuing*** | | |
| desired linear speed | 0.7 | $m\,s^{-1}$ |
| width of the plant model (in number of laser rays) | 15 | |
| height of the plant model | 0.3 | m |
| number of averaged Lidar scans | 25 | |
| minimal accepted distance between rows | 0.66 | m |
| maximal accepted distance between rows | 0.9 | m |
| proportional gain of the control law | 0.8 | |
| **Configuration of the switching algorithm** | | |
| minimal correlation value to transit to *furrow* | 0.4 | m |
| maximal correlation value to transit to *path* | 0.2 | m |
| time to transit to *furrow* | 0.5 | s |
| time to transit to *path* | 0.3 | s |

The results of the experiment are presented in Figure 10. In Figure 10a, the trajectory taken by the robot is in red when it uses *path tracking* and in blue when it uses *furrow pursuing*. The reference path for *path tracking* is black but not often visible. The lateral

errors of both control laws are displayed in Figure 10b using the same color code. The robot started its movement in the maize plot, made a complete loop and ended at the beginning of the same plot. Although the robot started in the maize plot, it used *path tracking* for a very short distance (about 0.7 m) since the plant detection algorithm needed to accumulate several Lidar measurements before being able to deduce a trajectory to follow. After that, the rest of the plot as well as the next plots were fully followed using the specialized behavior. However, we can see that the beginning and end of the phases using *furrow pursuing* are not well aligned, while in reality the plots are. Indeed, since the Lidar observed about 4 m ahead of the robot location point, the tracking of the plants would start and stop with the same advance. We can also notice that, although there was grass in the field between the plots, the program never switched to *furrow pursuing* during U-turns. This was because the few false plant detections did not occur continuously enough to switch to *furrow*.

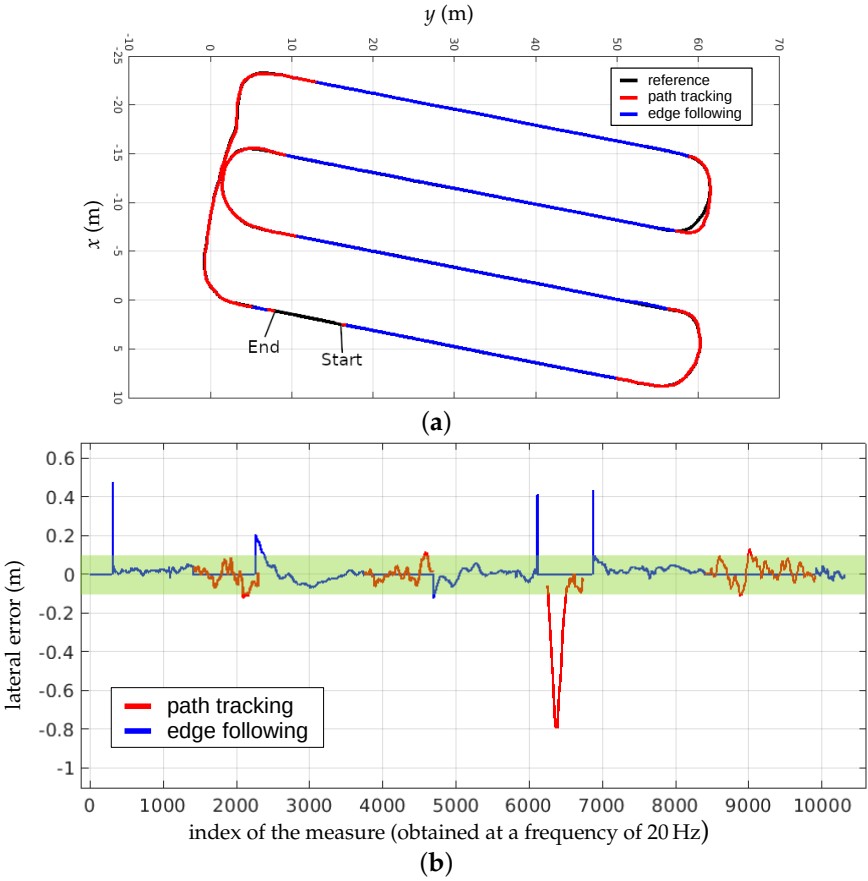

**Figure 10.** Results of experiment 2. (**a**) Absolute position $(x, y)$ of the robot. In red, the robot is in *path tracking* mode; in blue, it is in *furrow pursuing* mode. The black trajectory corresponds to reference trajectory to follow. (**b**) Lateral error in the Frénet frame of the local trajectory. For *path tracking* (in red), the local trajectory is the reference trajectory, and for *furrow pursuing* (in blue), it is built from the furrow detection.

The lateral errors measured over the whole experiment were relatively good and were often less than 0.1 m. It can be noticed that, contrary to the previous experiment, the phases using the specialized behavior were less noisy than the phases in *path tracking* mode. This means that the plant detection algorithm was able to generate a very stable trajectory. However, we can see that it caused some lateral error peaks at the beginning and the end of these phases because the linear regression representing the path to follow did not have enough points to give a correct direction. These errors were, however, not disturbing since the robot used the *path tracking* mode in these moments. There was also an

important deviation at the end of the third phase where the lateral error to the reference trajectory went down to −0.8 m. This comes from the fact that the robot finished the bean plot but the *furrow* behavior continued to wrongly detect plants for two meters. Since the reference trajectory had already started to turn left, the robot then fell behind its trajectory and traveled 6 m before recovering its error. This error was not critical since this was a plant-free zone, dedicated to turnaround maneuvers.

### 4.3. Experiment 3: Tracking Non Linear Ranks

This last experiment was similar to the previous one, as it was also a scenario involving bean and maize plots from the ACRE competition. This was the work conducted as part of the evaluation for field navigation. The scenario of this evaluation is presented in Figure 11a. It was made up as follows:

- A curved plot of beans followed by a curved plot of maize;
- A turning path;
- A maize plot followed by a bean plot on the same line;
- A turning path to return to the beginning.

The curvature of the first two plots was not very pronounced, but it allowed us to test if *furrow pursuing* was able to correctly follow trajectories that were not completely straight. The configurations of the different algorithms were similar to those of the previous experiment (Table 2).

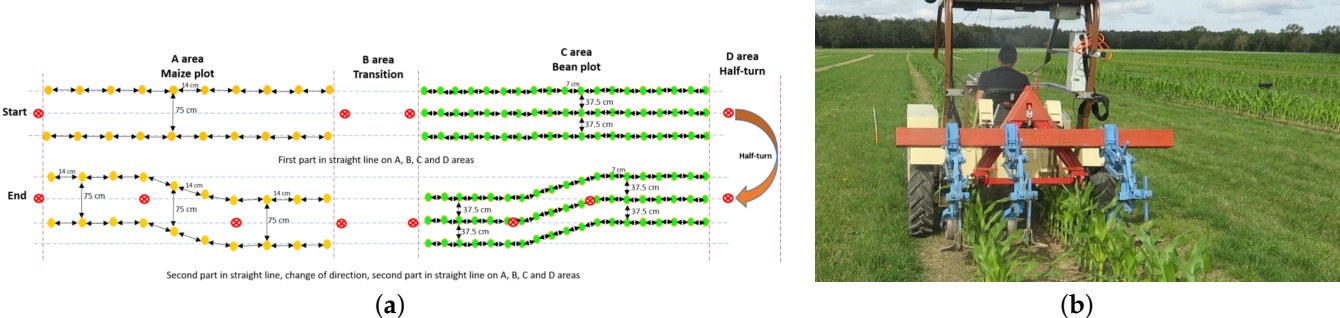

(a)      (b)

**Figure 11.** Presentation of experiment 2. (**a**) Scenario of the evaluation; (**b**) picture of the robot when it is following the curved maize plot.

The results are presented in Figures 12 and 13 using the same style as before. We can see that, globally, the reference trajectory and the plots were correctly followed with a lateral error of less than 0.1 m. There was still the problem of the lateral error peaks at the beginning and end of the phases in *furrow pursuing* mode, but they were still not taken into account by the switching algorithm. We can notice that the very slight curve present in the first two plots had no impact on *furrow pursuing*, although the detection algorithm approximated the rows of plants by straight lines. This was due to the fact that the detected trajectory was very short (about 4 m) and the curvature was not strong enough for the tracking error to be notifiable. We can also notice that there was a very short switching on *path tracking* before the end of the first plot. It was caused by missing plants in a part of this plot. Depending on the quality of the agricultural soil, some areas may be less suitable for plant development. For this reason, behaviors based on plant observation may fail. Our approach then has the advantage of being more robust and allowing us to continue to perform the task due to the redundancy of the control solutions.

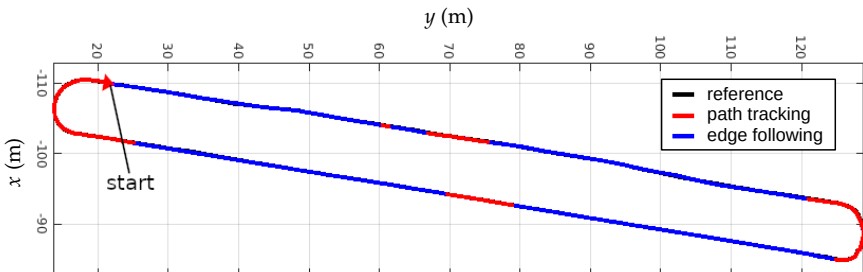

**Figure 12.** Absolute position $(x, y)$ of the robot. In red, the robot is in *path tracking* mode; in blue it is in *furrow pursuing* mode. The black trajectory corresponds to reference trajectory to follow.

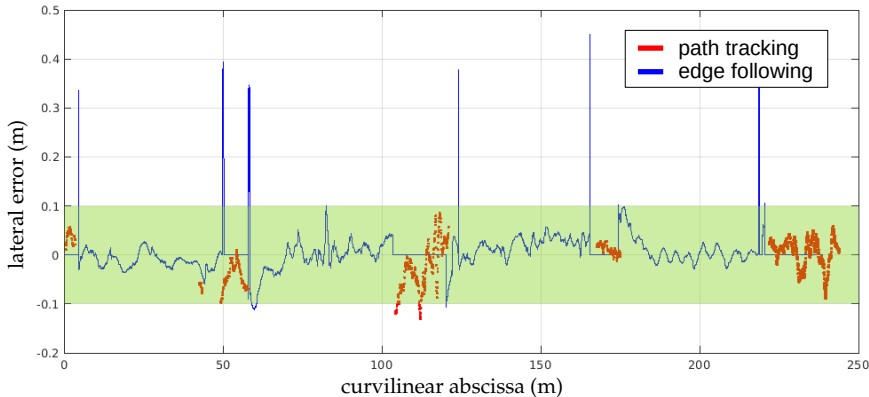

**Figure 13.** Lateral error in Frénet frame of the local trajectory. For *path tracking*, the local trajectory is the reference trajectory, and for *furrow pursuing*, it is built from the *furrow* detection.

## 5. Conclusions

In this paper, a global strategy to adapt the global behavior of an agricultural robot has been proposed by selecting the best approach depending on the context of evolution. Three basic behaviors have been here proposed: *path tracking*, *edge following* and *furrow pursuit*. Depending on the task to be performed and the result of perception, a criterion has been proposed to select in real time the most adapted control approach while avoiding the chattering effect and the use of a semantic map or preliminary mission planning, potentially complex to obtain or define. The effectiveness of the proposed approach has been tested through full-scale experiments on an automated electrical tractor on different field typologies. The first type of testing has been achieved in an orchard environment, using commutation between structure following and path tracking. The second series of tests has been performed on a open field, in order to actually remove weeds from maize and beans, using a dedicated implement. The results of these experiments show the ability to handle the complex robotics mission that is required to achieve a complete agricultural operation. The obtained accuracy, within a few centimeters with respect to actual rows of crops, matches the farmers' expectations to achieve agroecological tasks.

The proposed algorithm is compatible with the addition of new robotics behaviors without known limitations. Nevertheless, the extension to new behaviors (such as target tracking for cooperation) could imply extending the criteria. For the moment, such criteria only include the quality of crop or structure detection, without checking a good matching between the detected plants' structures and the planned trajectory. The use of such information is expected to be developed to allow the robot to switch to the trajectory tracking if false detection occurs or if the field limit is reached without a row completely ending.

In order to increase the robustness of the proposed algorithm, it would be also interesting to add plant recognition capabilities within the Lidar frame or by adding exteroceptive sensor such as a camera. Despite a strength of the switching approach being that it avoids the use of a semantic map, navigation based on a structure may lead the robot to follow any shapes looking at the expected one. As a result, for safety reasons or enhanced robustness,

the recognition of the nature of geometry would be an important element. This could be performed using reinforcement learning algorithms exploiting the database generated by current experiments. Additional information related to the plant and soil health could also be extracted, increasing the agroecological nature of the proposed developments.

**Author Contributions:** Conceptualization, C.P., R.L. and J.L.; methodology, C.P. and J.L.; software, C.P., J.L., V.R.; validation, R.L.; formal analysis, R.L.; investigation, C.P., J.L. and R.L.; resources, V.R.; writing—original draft preparation, C.P. and R.L.; writing—review and editing, C.P. All authors have read and agreed to the published version of the manuscript.

**Funding:** This work has been funded by the French National Research Agency (ANR) under the grant ANR-19-LCV2-0011, attributed to the joint laboratory Tiara (Available online: www6.inrae.fr/tiara (accessed on 31 July 2022)). It has also received the support of the French government research program "Investissements d'Avenir" through the IDEX-ISITE initiative 16-IDEX-0001 (CAP 20-25), the IMobS3 Laboratory of Excellence (ANR-10-LABX-16-01) and the RobotEx Equipment of Excellence (ANR-10-EQPX-44).

**Conflicts of Interest:** The authors declare no conflict of interest.

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
