# Peer review of "A Multi-Control Strategy to Achieve Autonomous Field Operation"

_agriengineering, doi:10.3390/agriengineering4030050_

Round 1

Reviewer 1 Report

In this paper, the authors propose a switching strategy between several perception and control approaches allowing the robot to realized accurately a whole mission. Three main behaviors are detailedly described in the framework of this manuscript, including path tracking, edge following, furrow pursuing. This manuscript is well written and has a certain value for the agroecology study.

A few suggestions to improve this manuscript:

Line 155: The full name of IMU and RTK-GNSS should be given when these abbreviations first appear in the manuscript.

Line 279: france -> France.

Figure 8: Please add the legend in subfigure (a) and (b).

Author Response

Thanks for your suggestions.

Line 155: The full name of IMU and RTK-GNSS should be given when these abbreviations first appear in the manuscript.

I have fixed it.

Line 279: france -> France.

I have fixed it.

Figure 8: Please add the legend in subfigure (a) and (b).

I have added legend to all result figures.

Reviewer 2 Report

General Comments: This paper proposes ‘a multi-control strategy to achieve autonomous field operation’. It can enable the robot to complete the overall control task. The path tracking method is used as the strategy basis. On this basis, the strategy design of edge tracking and ridge tracking is carried out. It has certain research significance and application value. However, the abstract part of the paper is redundant, the theoretical description is not detailed, the control strategy switching research part is incomplete and the experimental results are not fully described.

Specific comments:

1. In the abstract section, the preamble is too much, and the research content and method are not expressed in detail. The description of "accuracy is kept within a few centimeters" will lead to unclear summary of the experimental results, and the control strategy "has good robustness for different kinds of crops" is also proposed in the paper. "However, it is not reflected in the abstract.

2. In Chapter2, the theoretical description of path tracking is relatively clear, but the theoretical description of edge tracking and ridge tracking is not detailed enough. The author's intention can be understood in the review. The theoretical research of path tracking, edge tracking and ridge tracking is a progressive process, but the theoretical significance and difference of the three are not fully described and need to be further improved.

3. In Chapter 3, the method of switching the three control strategies is proposed, and the difficulty of this method is the selection of switching criteria between different behaviors. The "parameter criteria" mentioned in Chapter 3 is not specific, or the actual description can only be called "switching conditions". And how to realize the switching of the three is not explained in detail. Therefore, this part needs to be further improved. The parameter criteria are corresponding to the actual environment and the corresponding handover parameters are given.

4. In Chapter4, the test methods and contents of this article are relatively complete. However, the "high accuracy and good robustness" mentioned in the summary is not reflected. First, the accuracy of path tracking, edge tracking and ridge tracking is not clear in the case of no navigation and navigation. Secondly, although the application test of this strategy is carried out in the structural environment, it still needs to be tested in the terrain with obstacles and different slopes, so the robustness test needs to be improved. Finally, the switching speed and sensitivity of the three strategies should be tested to prove the robustness of the overall strategy.

Author Response

Thanks for your suggestions.

1. In the abstract section, the preamble is too much, and the research content and method are not expressed in detail. The description of "accuracy is kept within a few centimeters" will lead to unclear summary of the experimental results, and the control strategy "has good robustness for different kinds of crops" is also proposed in the paper. "However, it is not reflected in the abstract.

I have rewritten the beginning of the abstract to jump more quickly to the subject of this paper I have precised the robustness aspect.

2. In Chapter2, the theoretical description of path tracking is relatively clear, but the theoretical description of edge tracking and ridge tracking is not detailed enough. The author's intention can be understood in the review. The theoretical research of path tracking, edge tracking and ridge tracking is a progressive process, but the theoretical significance and difference of the three are not fully described and need to be further improved.

The theoretical aspect of edge tracking and ridge tracking are now more detailed by using mathematical notation and algorithms.

3. In Chapter 3, the method of switching the three control strategies is proposed, and the difficulty of this method is the selection of switching criteria between different behaviors. The "parameter criteria" mentioned in Chapter 3 is not specific, or the actual description can only be called "switching conditions". And how to realize the switching of the three is not explained in detail. Therefore, this part needs to be further improved. The parameter criteria are corresponding to the actual environment and the corresponding handover parameters are given.

To improve the understanding of the criteria, I have written the expression of its conditions and made a more direct link to its configuration in the experiments.

4. In Chapter4, the test methods and contents of this article are relatively complete. However, the "high accuracy and good robustness" mentioned in the summary is not reflected. First, the accuracy of path tracking, edge tracking and ridge tracking is not clear in the case of no navigation and navigation. Secondly, although the application test of this strategy is carried out in the structural environment, it still needs to be tested in the terrain with obstacles and different slopes, so the robustness test needs to be improved. Finally, the switching speed and sensitivity of the three strategies should be tested to prove the robustness of the overall strategy.

This paper can certainly be improved by making a more complex experiment scenario that implies some critical cases of the switching algorithm but it requires times and specific environments that will not be available in the restricted delay of this review.

Reviewer 3 Report

1. The Abstract should be revised. Half of it is background introduction. Also, descriptions of the work is unclear, e.g. “several” behaviors, “several” perception/control approaches, with “a good robustness” with respect to “different kinds” of crops, within “few” centimeters…

“The effectiveness of the proposed development is tested through full scale experiments in realistic field environment”, details of such a development should be added before this sentence.

2. Please consider adding information of the test platform/electrical tractor (mass, power, length, wheelbase, drivetrain, steering etc.) and of the key instruments used in the development (specifications of the GNSS, Lidar etc.)

3. There are two “proportional gain of the control law” in Table 1.

4. Please be consistent with the units used for the results ( m or cm).

5. What does “index of the measure” mean in Fig 10(b)? What’s the measurement frequency?

Author Response

Thanks for your suggestions.

1. The Abstract should be revised. Half of it is background introduction. Also, descriptions of the work is unclear, e.g. “several” behaviors, “several” perception/control approaches, with “a good robustness” with respect to “different kinds” of crops, within “few” centimeters…

“The effectiveness of the proposed development is tested through full scale experiments in realistic field environment”, details of such a development should be added before this sentence.

I have rewritten the beginning of the abstract to jump more quickly to the subject of this paper I have precised the unclear elements.

2. Please consider adding information of the test platform/electrical tractor (mass, power, length, wheelbase, drivetrain, steering etc.) and of the key instruments used in the development (specifications of the GNSS, Lidar etc.)

I have added this information at the beginning of the experiments chapter.

3. There are two “proportional gain of the control law” in Table 1.

I fixed it.

4. Please be consistent with the units used for the results ( m or cm).

I have choosed to express everything in meters.

5. What does “index of the measure” mean in Fig 10(b)? What’s the measurement frequency?

I have added more information to this legend to improve understanding of this figure.